Isolation and characterization of human articular chondrocytes from surgical waste after total knee arthroplasty (TKA)

Naranda Jakob jakob.naranda@gmail.com 1
Gradišnik Lidija 2
Gorenjak Mario 3
Vogrin Matjaž 1
Maver Uroš uros.maver@um.si 2
1 Department of Orthopaedics, University Medical Centre Maribor , Maribor , Slovenia
2 Institute of Biomedical Sciences, University of Maribor, Faculty of Medicine , Maribor , Slovenia
3 Center for Human Molecular Genetics and Pharmacogenomics, University of Maribor, Faculty of Medicine , Maribor , Slovenia
Abdala Virginia
Electronic publication date: 2017 Mar 21
Publication date: 2017
Volume: 5
Electronic Location ID: e3079
Received 2016 Oct 5; Accepted 2017 Feb 9
Copyright: ©2017 Naranda et al.
Copyright year: 2017
Copyright holder: Naranda et al.
License: This is an open access article distributed under the terms of the Creative Commons Attribution License, which permits unrestricted use, distribution, reproduction and adaptation in any medium and for any purpose provided that it is properly attributed. For attribution, the original author(s), title, publication source (PeerJ) and either DOI or URL of the article must be cited.
License URL: https://creativecommons.org/licenses/by/4.0/

Keywords: Human articular chondrocytes, Total knee arthroplasty, Isolation protocol, Phenotype preservation, Collagen 2, TKA, Gene expression, Aggrecan

Funding: Slovenian Research Agency P3-0036 I0-0029 University Medical Center Maribor, Slovenia IRP 2012/02-11 This work was supported by the Slovenian Research Agency (grant numbers: P3-0036 and I0-0029), as well by the University Medical Center Maribor, Slovenia (IRP 2012/02-11). The funders had no role in study design, data collection and analysis, decision to publish, or preparation of the manuscript.

==============================
Background

Cartilage tissue engineering is a fast-evolving field of biomedical engineering, in which the chondrocytes represent the most commonly used cell type. Since research in tissue engineering always consumes a lot of cells, simple and cheap isolation methods could form a powerful basis to boost such studies and enable their faster progress to the clinics. Isolated chondrocytes can be used for autologous chondrocyte implantation in cartilage repair, and are the base for valuable models to investigate cartilage phenotype preservation, as well as enable studies of molecular features, nature and scales of cellular responses to alterations in the cartilage tissue.

Methods

Isolation and consequent cultivation of primary human adult articular chondrocytes from the surgical waste obtained during total knee arthroplasty (TKA) was performed. To evaluate the chondrogenic potential of the isolated cells, gene expression of collagen type 2 (COL2), collagen 1 (COL1) and aggrecan (ACAN) was evaluated. Immunocytochemical staining of all mentioned proteins was performed to evaluate chondrocyte specific production.

Results

Cartilage specific gene expression of COL2 and ACAN has been shown that the proposed protocol leads to isolation of cells with a high chondrogenic potential, possibly even specific phenotype preservation up to the second passage. COL1 expression has confirmed the tendency of the isolated cells dedifferentiation into a fibroblast-like phenotype already in the second passage, which confirms previous findings that higher passages should be used with care in cartilage tissue engineering. To evaluate the effectiveness of our approach, immunocytochemical staining of the evaluated chondrocyte specific products was performed as well.

Discussion

In this study, we developed a protocol for isolation and consequent cultivation of primary human adult articular chondrocytes with the desired phenotype from the surgical waste obtained during TKA. TKA is a common and very frequently performed orthopaedic surgery during which both femoral condyles are removed. The latter present the ideal source for a simple and relatively cheap isolation of chondrocytes as was confirmed in our study.

Introduction

Damage to articular cartilage has important clinical implications since the cartilage tissue possesses a limited intrinsic healing potential and tends to an incomplete regeneration by local chondrocytes, accompanied with an inferior fibrocartilage formation (Camp, Stuart & Krych, 2014; McNickle, Provencher & Cole, 2008; Richter et al., 2016). Surgical intervention is often the only option, although the repair of damaged cartilage is often less than satisfactory, and rarely restores full function or returns the tissue to its native state (Kerker, Leo & Sgaglione, 2008; Kock, Van Donkelaar & Ito, 2012; Tuli, Li & Tuan, 2003). Over the past decade a number of viable options of cartilage regeneration have been introduced into clinical practice (Camarero-Espinosa et al., 2016; Hettrich, Crawford & Rodeo, 2008; Schrobback et al., 2011). Among these, autologous chondrocyte implantation (ACI) seems the most promising since it relies on the use of biodegradable materials that serve as temporary cell-carriers, enabling in vitro cell growth and subsequent implantation into the defective cartilage (Bomer et al., 2016; Niemeyer et al., 2016; Robb et al., 2012).

Tissue engineering of articular cartilage remains challenging due to the specific structure of cartilage tissue, i.e., its multiphasic cellular architecture together with remarkable weight-bearing characteristics (e.g., resistance to mechanical stress and wear) (Kim, Shin & Lim, 2012; Su et al., 2012). Good understanding of the cartilage structure, physiology, and the molecular basis of chondrogenesis is key to in vitro cartilage production, either for use in tissue engineering or clinics (Bhat, Tripathi & Kumar, 2011; Lee et al., 2013; Li et al., 2012). The state-of-the-art concept of in vitro cartilage tissue development combines the use of biocompatible and biodegradable carrier materials, the application of growth factors, the use of different cell types (stem or already differentiated) and different approaches to simulate the native mechanical stimulation (Gardner et al., 2013; Hildner et al., 2011; Khan et al., 2013; Naranda et al., 2016).

More specific challenges of articular cartilage tissue engineering remain the high consumption of cells and related costs, as well as the preparation of an ideal host scaffold. Although solutions to both mentioned challenges have been introduced in recent years (Bassleer, Rovati & Franchimont, 1998; Stellavato et al., 2016), the cell part is gaining far less research momentum. Therefore, it comes to no surprise that novel approaches for chondrocyte isolation are highly desired, especially considering the high prices of ordered cells. Optimisation of isolation yields, abundant cell sources and efficient culturing procedures that lead to preparation of desired, reproducible and relatively affordable cell cultures or/and material-cell constructs with good durability are therefore highly rated novelties in recent research (Dehne et al., 2009; Naranda et al., 2016; Otero et al., 2012).

Several methods for chondrocyte isolation from various tissue parts and organisms were introduced over the last decades (Hu et al., 2002; Li et al., 2015; Mirando et al., 2014; Shortkroff & Spector, 1999; Strzelczyk, Benke & Gorecki, 2001; Xu & Zhang, 2014). Although their cell source varies, the crucial steps of these reported isolation protocols have a lot of common ground. One of the main similarities to digest the harvested tissue during the preparation of the primary culture is the use the enzyme type 2 collagenase (Hayman et al., 2006; Lagana et al., 2014). Variations in the time of the tissue exposure to the enzyme (Hayman et al., 2006), as well as combining it with other enzymes (trypsin, pronase, hyaluronidase etc.) is not unusual (Jakob et al., 2001). Several examples of effective chondrocyte isolation procedures including the source tissue and organism, the digestion enzyme, time of tissue exposure and the cell yield, were summarized by Oseni, Butler & Seifalian (2013). In their study, Oseni, Butler & Seifalian (2013) evaluated the necessary isolation and characterization procedures that would give a maximum yield with optimal cell viability for the engineering of large cartilaginous constructs such as the human nose and ear. At this point it is important to mention that the state of the source tissue has also to be accounted for Lagana et al. (2014). In this context, Lagana et al. performed characterization of basic parameters of articular chondrocytes isolated from 211 osteoarthritic patients. They concluded that a systematic characterization of the cellular yield and chondrocyte proliferation rates is very useful in view of a possible autologous cell therapy (Lagana et al., 2014). Therefore, it is very important to determine the quality of the cell source, which is known to greatly influence the outcome of engineered tissue (Lagana et al., 2014).

The most demanding part in the process of in vitro culturing still presents the preservation of the desired phenotype to a high enough passage to yield sufficient cells to perform planned experiments (Pei & He, 2012; Rosenzweig, Solar-Cafaggi & Quinn, 2012; Schnabel et al., 2002; Wu et al., 2014). Since the latter depends on numerous factors and can therefore be confirmed only by a combination of (often) expensive techniques (different microscopies, molecular analysis, immunocytochemistry etc.), it is important to prepare protocols for an easier and cheaper preliminary phenotype confirmation by means of methods, available in most cell laboratories around the world. Since the desired phenotype can be identified by chondrocyte specific production (Chen et al., 2014; Han et al., 2010), we believe that the easiest and safest preliminary method to prove phenotype preservation could be the analysis of gene expression. More specifically, this analysis should include the evaluation expression of genes related to cartilage specific markers (e.g., collagen type 2 and aggrecan). To follow-up possible dedifferentiation towards the fibroblastic phenotype (Duan et al., 2015; Goldring, Tsuchimochi & Ijiri, 2006; Haudenschild et al., 2001; Makris et al., 2015; Otero et al., 2012), we propose simultaneous measurement of up-regulation of collagen type 1.

Based on all mentioned it is clear that chondrocyte isolation from an abundant source with a high yield, together with an effective and cheap preliminary phenotype confirmation method, would be greatly beneficial to boost the development of cartilage tissue engineering (Cetinkaya et al., 2011; Goepfert et al., 2010; Schrobback et al., 2011). This study was therefore designed to provide a relatively simple, yet effective procedure for isolation and culturing of human tissue derived primary chondrocytes up to the second passage. As the preliminary method of phenotype confirmation, we chose the evaluation of chondrocyte specific gene expression, together with morphological evaluation of cells. Such an approach provides a cheap and effective protocol to be considered an alternative to other available methods (Hu et al., 2002; Li et al., 2015; Strzelczyk, Benke & Gorecki, 2001; Xu & Zhang, 2014), especially suitable for other laboratories to boost their respective entry level cartilage tissue engineering studies. To confirm our claims and the overall effectiveness of the used approach, immunocytochemical analysis of the most important chondrocyte specific extracellular matrix products (aggrecan and collagen type 2) were evaluated after one week of cell growth (for the second passage). To observe the tendency of the chondrocyte cells towards differentiation into fibroblast like cells, collagen type 1 was also evaluated using the same approach.

Materials and Methods

Materials

All used materials and chemicals were of laboratory grade and purchased from Sigma-Aldrich, Germany, if not stated otherwise. For specific parts of the isolation process and cultivation, all used labware and chemicals were additionally sterilized using the standard autoclavation procedure (Avtoklav A-21, Kambič, Slovenia).

Isolation of primary chondrocytes

Full-thickness cartilage was surgically removed from the femoral condyle of arthritic knee of a 50 years old patient who underwent total knee arthroplasty (TKA) performed at the University Medical Centre Maribor, Slovenia (application reference: 123/05/14). Prior to surgery, no systemic disease or any treatment was reported for the donor patient. The study was conducted in accordance with the Declaration of Helsinki and its subsequent amendments and was approved by the Republic of Slovenia National Medical Ethics Committee (Ljubljana, Slovenia). The patients’ written consent was obtained.

The cartilage tissue was surgically removed under sterile conditions during TKA procedure. The standard cutting blocs for femoral resection were used and resection was performed in the usual manner. Distal and/or posterior femoral condyles were used for chondrocyte isolation depending on the macroscopic condition of the cartilage tissue (due to e.g., osteoarthritis). Immediately after the removal, the bone cuts were transferred into a previously sterilized 250 ml glass bottle filled with phosphate buffered saline (PBS; Sigma-Aldrich, Munich, Germany) and immediately brought to the cell isolation laboratory.

The cartilage-bone tissue was transferred to a petri dish filled with PBS to prevent drying of the tissue. In the cell isolation laboratory, the cartilage tissue was carefully removed from the bone cuts surface using a No 11 blade to obtain approximately 2 x 2 mm pieces of cartilage tissue. PBS was carefully removed by a pipette and the petri dish was immediately filled with 10 mL solution of 0.25 wt.% Trypsin/EDTA (Sigma, France). The as-prepared cartilage pieces were incubated for 3 h at 37°C and 5 wt.% CO2 (CO2 Incubator MCO-19AICUVH-PE; Panasonic, Tokyo, Japan), followed by addition of 20 mL of Advanced Dulbecco’s modified Eagle’s medium (Advanced DMEM; Gibco, Grand Island, NY, USA) to the cell suspension. The suspension was transferred to a 50 mL falcon tube and centrifuged at 300 x g for 10 min (Centrifuge 5804 R; Eppendorf, Hamburg, Germany). The supernatant was carefully discarded and the cell pellet was re-suspended in 20 mL of Advanced DMEM and centrifuged at 200× g for 5 min (Centrifuge 5804 R; Eppendorf, Hamburg, Germany). The supernatant was again carefully discarded and the cell pellet re-suspended in 10 mL Advanced DMEM supplemented with 100 IU/ml Penicillin, 1 mg/ml Streptomycin, 2mM L-glutamine and 5 wt.% foetal bovine serum (FBS; Gibco, Grand Island, NY, USA) and plated on 25 cm2 flasks (in triplicates). In the cell pellet, very small fragments of cartilage were also present. Besides primary chondrocytes, these fragments were also seeded and after a week of incubation, the cells were observed crawling from the tissue fragments. Together with the primary chondrocytes these were then left until confluence was reached.

Growing cells were regularly observed with an Axiovert 40 inverted optical microscope (Zeiss, Oberkochen, Germany) at several magnifications. The culturing medium was changed every three days. The general steps of the procedure are schematically depicted in Fig. 1.

Figure 1 Chondrocyte isolation from cartilage in a short overview of the most important preparation steps.

Gene expression analysis

Gene expression analysis of cartilage specific markers collagen type 2 (COL2) and aggrecan (ACAN) was performed in order to determine the primary chondrogenic phenotype. Possible dedifferentiation to a more fibroblast like cell type was evaluated by monitoring the expression of collagen type 1 (COL1). After confluence was reached in all respective samples (triplicates) (see above ‘Isolation of primary chondrocytes’. for details), the cell suspension was transferred to micro-centrifuge tubes, and 1.4 mL of TRI reagent (Sigma-Aldrich, Munich, Germany) was added. The tubes were vortexed for 30 min at room temperature. Afterwards, 280 µL of chloroform (Sigma-Aldrich, Munich, Germany) was added and the tubes were further vortexed for 15 min and centrifuged at 12.000 rpm and 4°C. RNA extraction was carried out according to the manufacturer’s instructions (Chomczynski, 1993; Louveau, Chaudhuri & Etherton, 1991). Concentration and purity of the extracted cellular RNA was determined using NanoDrop 2000c (Thermo Scientific, Waltham, MA, USA) through optical density readings at 260 nm and a 260/280 nm ratio. cDNA was obtained by using a cDNA reverse transcription kit (Applied Biosystems, California, USA). Primer sequences for cartilage target genes ACAN and COL2 were obtained from Caterson et al. (2001), while the corresponding mRNA sequences were retrieved from PubMed Nucleotide database (http://www.ncbi.nlm.nih.gov/nuccore/) and the AceView database (Thierry-Mieg & Thierry-Mieg, 2006). Primers for the target gene COL1 were designed using IDT oligo analyser (http://eu.idtdna.com/calc/analyzer). The primer sequences with the corresponding mRNA sequences and the corresponding NCBI accession numbers are given in Table 1. 2 µL of each cDNA sample with concentration of 15 ng/µL was used for quantitative real time PCR (qPCR) analysis performed using LightCycler 480 thermocycler (Roche, Switzerland) and with 2 × Maxima SYBR Green qPCR master mix (Life Technologies, Carlsbad, CA, USA) according to the manufacturer’s instructions. The quality and specificity of PCR amplicons were checked using melting curve analyses and agarose gel electrophoresis. All shown results are presented as average values with the standard errors.

Table 1 Primer sequences with the corresponding mRNA sequence and the corresponding NCBI accession numbers.

Gene	Gene name	Accession number	Primer sequence 5′ → 3′	
ACAN	Aggrecan	NM_013227.3	TGAGGAGGGCTGGAACAAGTACC	
	NM_001135.3	GGAGGTGGTAATTGCAGGGAACA	
COL1	Collagen type 1, alpha 1	NM_000088.3	CGGCTCCTGCTCCTCTTAG	
		CACACGTCTCGGTCATGGTA	
COL2	Collagen type 2, alpha 1	NM_001844.4	TTTCCCAGGTCAAGATGGTC	
	NM_033150.2	CTGCAGCACCTGTCTCACCA	
GAPDH	Glyceraldehyde-3-phosphate dehydrogenase	NM_001289745.1	GGGCTGCTTTTAACTCTGGT	
	NM_002046.5	TGGCAGGTTTTTCTAGACGG	
	NM_001289746.1		
	NM_001256799.2		

Immunocytochemistry

We characterized cells according to the expression of specific surface proteins (COL1, COL2, ACAN). Additional staining was performed in order to analyse the cells’ general morphology (cytoskeleton (actin)—using Phalloidin—iFluor 555 Reagent (Abcam, Cambridge, UK); nucleus—using mounting medium with 4′,6-diamidino-2-phenylindole (DAPI; Sigma-Aldrich, Munich, Germany)). Some more details about respective methods are described below. All micrographs were taken using either Floid Cell Imaging Station (Thermo Fisher Scientific, Waltham, MA, USA) or EVOS FL Cell Imaging System (Thermo Fisher Scientific, Waltham, MA, USA).

General protocol for immunocytochemistry

Round glass slides (2r = 12 mm) were placed on the bottom of wells in a P24 plate (in triplicate for each used dye) similar to the procedure used by Oseni, Butler & Seifalian (2013). Isolated cells (from the second passage) at a density of 50,000 cells / well were placed on each of the glass slides and incubated at 37°C, 5 wt.% CO2 for seven days. The medium (Advanced DMEM, supplemented with foetal bovine serum (FBS, Gibco, Grand Island, NY, USA)) was removed and the cells were washed with phosphate buffered saline (PBS; Sigma-Aldrich, Munich, Germany) once. Fixation of cells was performed using the Fixation Solution (Millipore, Billerica, MA, USA) for 10 min at room temperature, followed by washing of the cells three times with cold PBS (∼4°C).

Further sample handling differed for respective staining procedures. Namely, ACAN, COL1 and COL2 were stained using primary and secondary antibodies (the manufacturers protocols were followed for this purpose), whereas actin was stained in a single step (again, according to the manufacturers protocol).

Actin staining

Following the general protocol for immunocytochemistry, the working solution of the conjugated Phalloidin (1,000× Phalloidin stock solution in dimethyl sulfoxide DMSO (Abcam, UK), 1/1,000 dilution in PBS with 1 wt.% bovine serum albumin (BSA; Sigma-Aldrich, Munich, Germany) and 0.1 wt.% Tween 20 (Sigma-Aldrich, Munich, Germany)) was added. Incubation was performed for 90 min at room temperature and in a dark room. Rinsing was performed with PBS and was repeated three times. The final step was the addition of the Fluoroshield Mounting Medium with DAPI. Micrographs were taken at the suitable wavelengths for respective used dyes (excitation/emission: DAPI = 306/460 nm and Phalloidin = 556/574 nm).

Staining of COL1, COL2 and ACAN

Following the general protocol for immunocytochemistry described above, the cells were incubated for 30 min with PBS, supplemented with 1 wt.% BSA and 0.1 wt.% solution of Tween 20 to block nonspecific binding of antibodies. All incubations with the primary antibodies was performed overnight at 4∘C. Respective dilutions (in PBS with 1 wt.% BSA and 0.1 wt.% solution of Tween 20) of the primary antibodies were as follows:

1. ACAN: Anti-Aggrecan antibody [6-B-4] (Abcam, UK), 1:50,

2. COL2: Anti-Collagen 2 antibody (Abcam, UK), 1: 200,

3. COL1: Anti-Collagen 1 antibody (Abcam, UK), 1: 500.

After incubation, the cells were washed three times with PBS for 5 min. Incubation of cells with the secondary antibodies was performed in a dark at room temperature for 1 h (the same procedure was used also as the control for the attachment of respective secondary antibodies). The dilutions of the secondary antibodies (in PBS with 1 wt.% BSA and 0.1 wt.% solution of Tween 20) were as follows:

1. ACAN: Rabbit Anti-Mouse IgG H & L (Alexa Fluor 488) preabsorbed (Abcam, UK), 1: 1,000,

2. COL2 and COL1: Goat anti-rabbit IgG H & L (Alexa Fluor 594) (Abcam, UK), 1: 1,000.

After incubation, the cells were washed three times with PBS for 5 min. Finally, three drops of the Mounting Medium Fluoroshield with DAPI were added and the solution was left on the cells for 5 min. Micrographs were taken at the suitable wavelengths for respective used dyes (excitation/emission: ACAN = 495/519 nm and COL2/COL1 = 590/617 nm).

Results

Isolation of primary chondrocytes

As mentioned in the Materials and methods section, the full-thickness cartilage was obtained from the femoral condyle of an arthritic knee during knee arthroplasty (TKA) performed at the University Medical Centre Maribor, Slovenia. TKA is a common procedure at the mentioned hospital, considering that approximately 700 such surgeries are performed each year (Univerzitetni klinicni Center, 2014). Since the removed cartilage tissue is considered surgical waste, this presents a reliable and continuous source for isolation of primary chondrocytes.

The primary chondrocytes were isolated as described in the Materials and methods section. During their cultivation, their morphology and proliferation were regularly observed using inverted optical microscopy (Fig. 2). Figure 2A shows the thin slice of cartilage that was used for their cultivation, while Figs. 2B–2D present the primary human chondrocytes in a monolayer culture at different cultivation times. This initial examination was performed to follow possible morphological changes in the cell shapes, which would indicate possible dedifferentiation.

Figure 2 (A) Thin slice of cartilage for primary chondrocyte isolation; (B–D) the primary human chondrocyte culture in a monolayer after 3, 6 and 9 days, respectively.

The magnification of all shown images is 50×.

The full confluence of the isolated cells for the first and second passage was reached after two (14 days) and after one (seven days) week, respectively. Cell growth stopped presumably due to contact inhibition (Lackie, 2013). A comparison between the primary chondrocyte culture and the obtained chondrocyte cultures after the first and second passages is shown in Fig. 3. The cells formed confluent monolayers (after the above mentioned cultivation times) and appeared polygonal in shape (Figs. 3A–3C). It can be observed that the chondrocyte morphology became more spindle-like in the second passage (Fig. 3C), showing their tendency for dedifferentiation, most likely towards fibroblast-like cells (Hong & Reddi, 2013). Observing the mentioned changes was an indication that the third passage will not yield a high percentage of chondrocytes only using the proposed cultivation conditions.

Figure 3 Human chondrocyte culture: (A) the explant culture of chondrocytes (“primary culture”), (B) monolayer of chondrocytes after first passage, and (C) monolayer of chondrocytes after the second passage.

The magnification of all shown images is 50× (the inlay images were taken with a magnification of 100×).

Gene expression analysis of the isolated chondrocytes

Now that we determined the suitable number of passages presumably yielding a high percentage of chondrocyte cells, we performed additional characterization to confirm the chondrocytes’ desired phenotype. Analysis of gene expression was chosen due to its affordability and availability. The isolated cells from the human articular cartilage were characterized in regard of the genes related to specific chondrogenic production, namely collagen type 2 (COL2) and aggrecan (ACAN). To detect possible dedifferentiation towards fibroblast like cells, expression of collagen type 1 (COL1) was also determined. Expression of all three mentioned genes was performed after the confluence was reached for the second passage (after seven days). As shown in Figs. 4 and 5, both cartilage specific genes (COL2 and ACAN) and also the marker of fibrocartilage (COL1) were expressed in the isolated chondrocytes in both passages. qPCR results are presented as absolute Ct values. Reference gene GAPDH was used as an internal control (Chen et al., 2016).

Figure 4 cDNA products of analysed genes (GAPDH, collagen type 1, collagen type 2 and aggrecan) at the end-point of qPCR on agarose gel electrophoresis.

Analyzed genes: GAPDH (702 bp), COL2 (377 bp), COL1 (137 bp), ACAN (350 bp) and DNA markers (433 bp, 245 bp, 203 bp, 114 bp).

Figure 5 Results of qPCR analysis presented as absolute Ct values of target genes expression (ACAN, COL1, COL2 and GADPH).

The results are presented as average values with the standard errors of a triplicate.

Immunocytochemistry

We performed immunocytochemistry on the isolated cells to investigate chondrocyte phenotype alterations (ACAN, COL1 and COL2). Additionally, the cytoskeleton (actin) and cell nucleus were stained to show the overall healthy morphology of the cells. All staining was performed in three repetitions. As the negative control, staining only with the respective secondary antibodies, as well as with the Mounting medium with DAPI (after one day and after two days), were used. Production of all three proteins was confirmed (Figs. 6A–6C), which is in agreement with the results from the molecular analysis. All negative controls have shown no fluorescence, confirming the effectiveness and specificity of the used protocols. Staining of actin (Fig. 6D) confirmed the expected morphology of healthy cells, which is in agreement with the micrographs using optical microscopy.

Figure 6 Micrographs of the stained samples: (A) for ACAN, (B) for COL2 and (C) COL1. Additionally, (D) shows the cells with a stained cytoskeleton (actin).

For all samples a mounting medium with DAPI was used to stain the nuclei. The magnification of all shown images is 460× (according to the manufacturers microscope specifications).

Discussion

The development of novel solutions related to any tissue engineering application consumes a huge number of cells to prove safety and efficiency (Groeber et al., 2012; Maver et al., 2015; Mohd Hilmi & Halim, 2015; Rodriguez-Vazquez et al., 2015). Cartilage tissue engineering is no exception, and hence large scale expansion of chondrocytes is required either for novel scaffold testing, determination of potential cytotoxic effects of medical devices and implants for orthopaedic use (Bomer et al., 2016; Camarero-Espinosa et al., 2016; Makris et al., 2015). Cultivation of such high cell counts is a demanding task, especially considering the low number of obtained cells in the primary culture, and an often limited amount of available tissue. Consequently, further expansion and consecutive passages are needed, which on the other hand can lead to dedifferentiation (Mirando et al., 2014; Shortkroff & Spector, 1999; Thirion & Berenbaum, 2004). The latter is evident by morphological changes of the cells from polygonal to more elongated, as well as through a reduction in the growth rate (Cetinkaya et al., 2011; Haudenschild et al., 2001; Otero et al., 2012). For example, development of novel scaffolds for cartilage tissue engineering often requires a million cells per sample scaffold (the number depends on the size of the scaffold to be tested), exposing the high demand for cells and at the same time one of the major bottlenecks in development of novel tissue engineering solutions. At later passages, the quality of chondrocytes gradually decreases and is characterized with many of the phenotypic traits of fibroblast like cells and an increased synthesis of collagen type 1, rather than type 2 (Bonaventure et al., 1994; Diekman et al., 2010; Schnabel et al., 2002).

A sufficient number of cells can be ensured either through significant expenses (purchase of cells from different cell banks) or isolation of desired cells from tissues. While the first scenario requires sufficient funds, the latter requires appropriate tissue sources, an approval of respective Committees of Medical Ethics, and a rigorous final analysis to confirm the isolation of the desired cell type only. Since we work in the close proximity and in tight collaboration with the local University Medical Centre, the second scenario was more convenient. Our goal was to prepare a simpler and generally available protocol, which would include the isolation of primary chondrocytes from full-thickness cartilage that is surgically removed from the femoral condyle of an arthritic knee during total knee arthroplasty (TKA). As a preliminary prove of the protocols’ efficiency, we considered gene expression analysis as the best option, since it is affordable and the required instrumentation (PCR, inverted optical microscope) is most likely available in most cell biology laboratories. The set of analysed genes was carefully chosen considering the available literature to monitor cartilage phenotype alterations (Caterson et al., 2001; Diekman et al., 2010; Grogan et al., 2014; Jonitz et al., 2012; Seda Tigli et al., 2009; Shi et al., 2014). Based on the mentioned, the correlation between COL2 and COL1 in addition to ACAN, seemed to be the most suitable. For confirmation of the effectiveness of the proposed approach in terms of chondrocyte specific production besides the gene expression, immunocytochemical staining of COL2, COL1 and ACAN was used as well. The latter confirmed the chondrocyte specific productions (ACAN and COL2), as well as the presence of COL1, which could be an indication of ongoing dedifferentiation to more fibroblast like cells.

In general, the chondrocyte isolation protocol can be divided into different stages: isolation, seeding and chondrocytes grow in culture, although description and number of steps can vary (Gosset et al., 2008; Thirion & Berenbaum, 2004). After initial plating of the primary cultures, the chondrocytes spread out after 2–3 days and after 4–7 days the sufficient amounts of total RNA may be extracted. Primary cartilage phenotype (often confirmed by evaluating the presence of COL2 and ACAN mRNAs) may be initially preserved, but the expression of nonspecific collagens (e.g., COL1) begins to appear already 7 days after isolation (Otero et al., 2005). Moreover, adult articular chondrocytes are strongly contact-inhibited and undergo a rapid change in phenotype and gene expression, termed “dedifferentiation”, when isolated from cartilage tissue and cultured on culturing plastics (Haudenschild et al., 2001). Therefore, primary chondrocyte cultures should be used for experimental analyses immediately before or just after confluence is reached to assure optimal matrix synthesis and cellular responsiveness (Schneevoigt et al., 2016).

In the last two decades, several chondrocyte isolation protocols were developed and reported on (Hayman et al., 2006; Hu et al., 2002; Jakob et al., 2001; Lagana et al., 2014; Oseni, Butler & Seifalian, 2013; Strzelczyk, Benke & Gorecki, 2001). For example, an important recent study was conducted by Lagana et al. (2014), who isolated chondrocytes from 211 osteoarthritic (OA) patients undergoing total joint replacement. The authors of this study analysed specific features of chondrocytes such as cellular yield, cell doubling rate and the dependence between these parameters and patient-related data (e.g., joint type, age and gender). They concluded that such a systematic characterization of important cell source parameters could be useful in view of a possible autologous cell therapy for osteoarthritis, since the cell source quality is known to greatly influence the outcome of engineered tissue (Lagana et al., 2014). Another crucial study that we studied in details prior to our experimental design, was performed by Oseni, Butler & Seifalian (2013). In this study, the authors focused on a very important factor related to possible clinical use of cartilage tissue engineered products, namely the optimization of the isolation protocol to allow for a large-scale production. The result of their study was an optimized protocol with exactly defined isolation parameters (e.g., enzyme and concentration to be used, time of digestion and the seeding density for tissue culturing). Two other studies have to be mentioned in this context as well. Namely, the studies from Jakob et al. (2001) and Hayman et al. (2006), respectively. Jakob et al. focused on the research of possible chondrocyte isolation yield improvement by using various combinations of enzymes and reagents. Their results indicated that chondrocyte yields and capacity to attach and proliferate are not highly sensitive to the specific isolation protocol used (Jakob et al., 2001). Finally, Hayman et al. (2006) conducted a study, in which they tested combinations of three different enzymes and variable incubation/digestion times. A very important discussion point raised by the authors of this study was that different isolation protocols are to be used, if the focus is only on the yield or the goal is to produce preferentially “native” chondrocytes (Hayman et al., 2006).

The protocol of chondrocyte isolation described in this article led to successful growth and proliferation of cells with a proven chondrogenic potential up to the second passage as shown using molecular and immunocytochemical analysis. The characterization of primary human chondrocytes by molecular analysis showed the expression of cartilage specific genes (COL2 and ACAN), as well as a sign of dedifferentiation towards fibrocartilage for the second passage (indicated by the expression of COL1). In comparison with other available chondrocyte isolation protocols, we introduced some changes to the general protocol. As mentioned before, our target was a simple protocol with a high enough yield to conduct preliminary cartilage tissue engineering experiment, like testing of suitability of novel materials (Naranda et al., 2016). According to previous studies, the most commonly used enzyme in chondrocyte isolation, is type 2 collagenase (Hayman et al., 2006; Lagana et al., 2014; Oseni, Butler & Seifalian, 2013). Various incubation times are used to allow for tissue digestion, but in our experience, longer enzyme exposure times of tissues (and with longer exposures, an increasing number of cells as well) often lead to an increased number of dead cells and/or a lower yield of the cells with a desired phenotype. Considering all mentioned, we used a Trypsin/EDTA combination and an incubation time of 3 h. Although this is not the first research study reporting the use of trypsin for chondrocyte isolation (Hidvegi et al., 2006; Jakob et al., 2001), to the best of our knowledge it is the only one to use only this enzyme during the isolation protocol. Also, the reported incubation time is different to the mentioned studies. In addition, our protocol does not include the use of any growth factors like reported in some studies (Lagana et al., 2014), again with the focus to simplify the overall protocol. Moreover, no enzyme predigestion step was introduced in our protocol, like in some studies (Oseni, Butler & Seifalian, 2013).

The purpose of our study was not to revolutionize the chondrocyte isolation procedures, but rather to push the evolution of cartilage tissue engineering. As such, our desire was to present an alternative, affordable and relatively simple approach of chondrocyte isolation, especially suitable for laboratories working closely together with orthopaedic clinics. Such laboratories have the unique opportunity to use surgical waste materials, occurring during TKA. Since TKA is a very common surgery (considering the present demographics, the incidence will only increase (Peterson et al., 2015)), this approach could make cartilage-related studies far more available also for laboratories with limited resources, and hence push the overall development of this field towards novel and cheaper therapeutic solutions. Based on our results, we can claim that the combination of the use of surgical waste tissue occurring during TKA, and analysis by inverted optical microscopy and chondrocyte specific gene expression, as well confirmation of chondrocyte specific production, indeed results in an alternative and affordable means to boost cartilage-related research in the  future.

Conclusion

In this study, we describe a simple and affordable procedure of isolation and cultivation of human articular chondrocytes demonstrated a high chondrogenic potential to the second passage. As the source material, we propose the surgical waste tissue occurring during total knee arthroplasty (TKA). Chondrocyte cells are crucial not only for development of therapeutic approaches in cartilage repair (e.g., autologous chondrocyte implantation—ACI), but are necessary in cartilage tissue engineering to allow the development of functional cell models and novel scaffolds. For this purpose, chondrocytes have to be isolated in sufficient quantities and their phenotype should be preserved. Since all mentioned challenges are related to very high costs, we propose alternative isolation and testing protocols that are cheaper and could especially boost the preliminary studies related to cartilage research.

Additional Information and Declarations

Competing Interests

Author Contributions

Human Ethics

Data Availability

The authors declare there are no competing interests.

Jakob Naranda, Lidija Gradišnik and Mario Gorenjak conceived and designed the experiments, performed the experiments, analyzed the data, wrote the paper, prepared figures and/or tables, reviewed drafts of the paper.

Matjaž Vogrin and Uroš Maver conceived and designed the experiments, analyzed the data, contributed reagents/materials/analysis tools, wrote the paper, prepared figures and/or tables, reviewed drafts of the paper.

The following information was supplied relating to ethical approvals (i.e., approving body and any reference numbers):

The Republic of Slovenia National Medical Ethics Committee (Ljubljana, Slovenia) granted Ethical approval to carry out the study (application reference: 123/05/14).

The following information was supplied regarding data availability:

The raw data is located in the tables and figures in the manuscript.

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
