# Peer review of "Isolation and characterization of human articular chondrocytes from surgical waste after total knee arthroplasty (TKA)"

_PeerJ, doi:10.7717/peerj.3079_

## Round 0.1 · original submission · Major Revisions

After reading your letter in which you appealed my Reject decision, I have reconsider my first decision and I will give you the opportunity to respond to the criticisms raised by the reviewers via a rebuttal and a revised manuscript submitted to the system.

Please note that after you return your revised manuscript this may be send to the original and/or new reviewers for a new round of review.

· Appeal

Appeal

Following your decision in regard of our article, we have thoroughly reviewed the comments of the reviewers with my co-authors. We prepared a short argumentation in regard of the reasons for the rejection. Hope you can see it possible to review this argumentation and, based on the additional proposed changes in the manuscript, as well as the proposed additional experiments, reconsider your decision. Please find attached the argumentation.


· · Academic Editor

Reject

I have now received three reviews of your manuscript. I have carefully considered the opinions of our reviewers and think that your report of the gene expression without reference to non-chondrogenic cells and without comparison between different culture passages is a major flaw of the manuscript. Lack of quantitative data is also a major problem. Additionally, I am really concerned about the lack of chondricytes identification at the protein level with histological staining or other experiments.

Unfortunately, I think that the manuscript is not suitable for this journal.

Reviewer 1 ·

Basic reporting

The paper is written in a correct English and introduction and discussion are well referenced. The figures are relevant to the content of the article and of sufficient quality. However, insufficient and only qualitative results are described to support the study conclusions.

Experimental design

The scientific question has been properly defined but no reference to analogue existing methods have been provided, thus overstating the existing knowledge gap. The technical standard used to investigate the new procedure, even if described in sufficient detail, is quite poor.

Validity of the findings

No quantitative data are provided, at least on cell proliferation, which can support the conclusions that “the protocol…,yielding suitable number of cells” and “we describe an efficient… procedure”. Furthermore, only one patient has been used, thus not allowing to draw any statistically significant conclusion. Gene expression results are reported without reference to non-chondrogenic cells and without comparison between different culture passages, not allowing to conclude that chondrocytes “preserved chondrogenic phenotype”. When an alternative procedure is proposed, a proof should be given that the results obtained following the new method are better or at least not worse than those obtained with other procedures

Additional comments

Even if PeerJ does not judge papers about the novelty of data provided, the method proposed is a very standard procedure for the isolation of chondrocytes and thus reference to existing analogue protocols has to be mentioned in the introduction (e.g compare with Hayman DM et al. Tissue Eng. 2006 or Lagana et al. Cell Tissue Bank 2011). The rationale of “high pricing of ordered cells” or “since we are a small laboratory relying on limited available funds” could not be accepted as the main reasons to perform a scientific work, and thus similar expressions should be removed from the text

Reviewer 2 ·

Basic reporting

The English writing is acceptable.
The abstract did not conform to the PeerJ requirement.

Experimental design

There is insufficient novelty or significance as regard to the method depicted in the manuscript.

Validity of the findings

No Comments.

Additional comments

The manuscript introduced a method to isolate chondrocytes from waste of TKA and examined the characteristics. The English writing is acceptable. However, there is insufficient novelty or significance as regard to the method depicted in the manuscript.
1. Please give detailed discription of the slice culture and explant culture, including how the slice and explant were obtained, when and how the cells migrated out from the explant.
2. Do the authors use other enzymes except trypsin to digest the tissue? Only one sample may not be persuasive, repeated experiments are required.
3. Chondrocytes need to be identified at the protein level with histological staining or other experiments.
4. The authors need put more emphysis on the novelty in the manuscript of the method comparing with current methods.

·

Basic reporting

No comments

Experimental design

1. Chondrocytes are usually digested with type II collagenase. In this submission, chondrocytes are digested with 0.25% Trypsin/EDTA. Please add a procedure to compare the differences between these two methods.
2. Please set different time points , such as 1h, 3h, 6h, to compare the time effect of digestion on the cell viability.

Validity of the findings

Please discuss the reasons why you choose the digestion protocol with 0.25% Trypsin/EDTA instead of type II collagenase.

---

## Round 0.2 · accepted · Accept

Considering that you added immunostaining data and a detailed description of the culture method used, your work is ready to be published.

·

Basic reporting

No comments

Experimental design

Experimental design has been improved based on the reviewer's suggestions.

Validity of the findings

No comments.

Additional comments

The submission has been revised according to the reviewers' suggestions.

Reviewer 4 ·

Basic reporting

The English writing is acceptable and the introduction and discussion are well referenced.

Experimental design

The scientific question has been properly proposed and references have been supplemented. Detailed description of the culture method was provided. Immunostaining data have been provided.

Validity of the findings

no comments

Additional comments

no comments